# Conformation and Domain Movement Analysis of Human Matrix Metalloproteinase-2: Role of Associated Zn^2+^ and Ca^2+^ Ions

**DOI:** 10.3390/ijms20174194

**Published:** 2019-08-27

**Authors:** Leah Voit-Ostricki, Sándor Lovas, Charles R. Watts

**Affiliations:** 1Department of Neurosurgery, Mayo Clinic Health System-Franciscan Healthcare, La Crosse, WI 54601 USA; 2Department of Biomedical Sciences, Creighton University, Omaha, NE 68178, USA; 3Department of Neurologic Surgery, Mayo Clinic, Rochester, MN 55905, USA

**Keywords:** molecular dynamics, matrix metalloproteinase, domain movement, zinc binding protein, calcium binding protein

## Abstract

Matrix metaloproteinase-2 (MMP-2) is an extracellular Zn^2+^ protease specific to type I and IV collagens. Its expression is associated with several inflammatory, degenerative, and malignant diseases. Conformational properties, domain movements, and interactions between MMP-2 and its associated metal ions were characterized using a 1.0 µs molecular dynamics simulation. Dihedral principle component analysis revealed ten families of conformations with the greatest degree of variability occurring in the link region connecting the catalytic and hemopexin domains. Dynamic cross-correlation analysis indicated domain movements corresponding to the opening and closing of the hemopexin domain in relation to the fibronectin and catalytic domains facilitated by the link region. Interaction energies were calculated using the molecular mechanics Poisson Boltzman surface area-interaction entropy (MMPBSA-IE) analysis method and revealed strong binding energies for the catalytic Zn^2+^ ion 1, Ca^2+^ ion 1, and Ca^2+^ ion 3 with significant conformational stability at the binding sites of Zn^2+^ ion 1 and Ca^2+^ ion 1. Ca^2+^ ion 2 diffuses freely away from its crystallographically defined binding site. Zn^2+^ ion 2 plays a minor role in conformational stability of the catalytic domain while Ca^2+^ ion 3 is strongly attracted to the highly electronegative sidechains of the Asp residues around the central β-sheet core of the hemopexin domain; however, the interacting residue sidechain carboxyl groups are outside of Ca^2+^ ion 3′s coordination sphere.

## 1. Introduction

Matrix metaloproteinase-2 (MMP-2) is a 550 amino acid residue extracellular Zn^2+^ protease that degrades type I and IV collagens [1,2]. It is related to a family of 24 known endopeptidases with an active site Zn^2+^ ion. On the basis of evolutionary relationships and the structure of their domains, the family is divided into four subfamilies [3,4,5]. MMP-2 expression is associated with normal physiology, as well as several inflammatory, degenerative, and malignant diseases [6,7,8,9,10,11,12]. As shown in Figure 1; MMP-2 has three domains, catalytic (Cat), fibronectin (Fib), and hemopexin (Hpx) and five crystallographically assigned divalent cations (two Zn^2+^ and three Ca^2+^) [13]. The Cat domain (Tyr^110^ through Tyr^445^) is a conserved matrixin fold consisting of five β-sheets and three α-helices. The Fib domain (Glu^217^ through Gln^393^) is inserted within the catalytic domain between the β5-sheet and α2-helix and contains three type II fibronectin fingers consisting of two antiparallel β-sheets connected by a short α-helix forming a three prong treble hook-like arrangement. This domain and its arrangement may play a crucial role in substrate binding and presentation to the catalytic site. The Hpx domain (Leu^461^-Cys^660^) is a four bladed propeller fold that is partially oriented toward the catalytic domain. This domain binds an endogenous inhibitor TIMP-2; however, its role in enzymatic function is unknown. The Hpx and Cat domains are connected by a 16 amino acid proline rich Lnk region (Gly^446^ through Thr^460^) which is unresolved in the 1CK7 X-ray crystal structure and has been shown to be highly flexible in other molecular dynamics investigations [14,15].

Coordination geometries and interatomic metal cation to MMP-2 residue distances from the 1CK7 X-ray crystal structure are given in Table 1 [13,17,18]. The catalytic Zn^2+^ ion 1 is bound by the conserved MMP extended zinc binding motif [5]:HExxHxxGxxH⁄D,(1)
which in MMP-2 consists of interactions between the divalent cation and the sidechains of His^403^ and His^407^ from the α2-helix, and His^413^ from the Ω-loop (Figure 1B). The structural Zn^2+^ ion 2 is bound in a trigonal bipyramidal arrangement involving the sidechains of His^193^ and His^206^ from the β5- and β4-sheets respectively and Asp^180^ and His^178^ of the long S-loop of the Cat domain which is a conserved motif in the MMP family (Figure 1C). Two of the Ca^2+^ ions are bound near the Cat domain with Ca^2+^ ion 1 bound by the sidechains of Asp^208^ and Glu^211^ of the interim loop connecting the Cat and Fib domains and the carbonyl oxygens of Asp^185^, Gly^186^, Asp^188^, and Leu^190^ of the loop connecting the β3- and β5-sheets (Figure 1D). Ca^2+^ ion 2 is bound by the sidechains of Glu^166^ and carbonyl oxygens of Ala^167^ and Asp^168^ arising from the loop connecting the β1- and β3-sheets, the carbonyl oxygen of Gly^200^ of the loop connecting β5- and β4-sheets, and the sidechain of Asp^204^ arising from the β4-sheet, Figure 1E. The third structural Ca^2+^ ion 3, is bound by the carbonyl oxygens of Asp^476^, Asp^521^, Asp^569^, and Asp^618^ at the edge of the central cavity of the Hpx domain (Figure 1F).

The goal of the current study is to evaluate the dynamic stability of the divalent metal ions (2 Zn^2+^ and 3 Ca^2+^) reported in the X-ray crystal structure protein databank (PDB) ID: 1CK7 and examine the domain movements within MMP-2 using 1.0 μs NPT molecular dynamics (MD) simulations at physiological temperature (310 K). Protein-metal cation distances and MMPBSA-interaction entropy binding energies (ΔG) were calculated, and the sampled conformational space analyzed with dihedral Principle Component Analysis (dPCA) and Dynamic Cross-Correlation Matrix (DCCM) analysis. The stability of the divalent ions and domain movement conformations will play an important role in the development of an additive force field model for protein-ligand docking studies, subsequent dynamic protein-ligand simulations and potential MMP-2 inhibitor development.

## 2. Results and Discussion

### 2.1. System Equilibration and Conformational Stability

The Cα-trace configurational entropy of MMP-2 and associated divalent ions was calculated as a function of time (Appendix A) [19,20,21]. After a sharp rise in the configurational entropy over the first 100 ns, the value plateaus prior to 200 ns. Based on these results, we used the 200 ns to 1000 ns portion of the trajectory for our analysis with a sampling frequency of 0.1 ns. The radius of gyration (R_g_) and inter-domain center-of mass (COM) distances: Cat-Hpx and Fib-Hpx, were analyzed with k-means clustering and the associated means and standard deviations of each population calculated (Appendix A) [22,23]. Due to the large changes in R_g_ and COM distances between the domains and to ensure that the size of the solvation box was adequate, the minimum distances between periodic images as a function of simulation time were determined using the *g_mindist* utility in GROMACS (periodic minimum distance: 3.81 ± 0.75 nm). Five different distributions of protein conformations were identified with R_g_: 2.65 ± 0.02 nm, 2.70 ± 0.06 nm, 2.80 ± 0.05 nm, 2.83 ± 0.04 nm, and 3.09 ± 0.13 nm. The COM distances mirror the R_g_ results, identifying five distributions of Cat-Hpx COM distances: 3.55 ± 0.10 nm, 3.70 ± 0.17 nm, 3.79 ± 0.06 nm, 3.86 ± 0.13 nm, and 4.20 ± 0.23 nm. Five distributions of Fib-Hpx COM distances: 3.70 ± 0.06 nm, 3.84 ± 0.18 nm, 4.32 ± 0.22 nm, 4.48 ± 0.18 nm, and 5.52 ± 0.48 nm, were also identified. The R_g_ and COM distance data are consistent with inter-domain motions between Cat/Fib and Hpx and the presence of inter-domain motions and the sampling of more extended conformations of MMP-2 in solution compared to the more compact X-ray crystal structure (PDB ID: 1CK7) which has R_g_: 2.77 nm, Cat-Hpx COM distance: 3.81 nm, and Fib-Hpx COM distance: 2.00 nm [13]. These values are also consistent with those determined from prior simulations using the AMBER param03 force field [14,15]. It is not clear from either previous (multiple 200 ns simulations) or the current (1.0 μs) simulation if more extended conformations may be sampled or if an opening and closing of the Cat/Fib to Hpx domains occurs over longer time scales [14,15].

The average Cα-trace root mean square deviation (RMSD) data for MMP-2 as a whole and divided into its individual domains are given in Table 2. The most conformationally stable regions of the protein are the Cat and Hpx domains as observed in previous simulations with the majority of the Cα-trace flexibility accounted for within the three Fib domain, particularly the individual subdomains [14,15]. Although the RMSD of the Lnk region is low and comparable to that of Cat and Hpx, it should be considered that this is only a short 15 amino acid segment and that minor variations in the ϕ/ψ dihedral angles can result in dramatic changes in the relationships between the Cat and Hpx domains [15]. The Cα-trace root mean square fluctuation (RMSF) with the secondary structure assigned by the database of secondary structure assignments in the protein databank (DSSP) method is shown in Figure 2 [24]. Those regions with defined rigid secondary structure (β-sheets and α-helices) have lower RMSF values compared to more flexible β-turn/bend and coil regions. The greatest degree of conformational variability occurs within the Fib domain, while the most stable regions of the protein are within the Cat and Hpx domains. The stability of the Cat and Hpx domains is most likely secondary to the presence of the three long α-helices of the Cat domain and the prominent β-sheets and ordered arrangement of the Hpx domain. Although the ordered secondary structure is present within the Fib domain, it consists of three separate subdomains with a significant portion of the fold consisting of flexible β-turn/bend and coils. Flexibility within the treble hook arrangement of the Fib domain may play an important role in collagen binding and unraveling [25,26,27,28].

### 2.2. Conformational Analysis

The free energy landscape created by the first two dihedral principle components (dPC) is shown in Figure 3, with the corresponding lowest energy centroid conformations as determined by k-means clustering. The most conformationally stable region is the Cat domain in which the active site Zn^2+^ ion 1 is bound by His residues of the α1-helix and the Ω-loop (Figure 3 and Figure 4). The structure around Ca^2+^ ion 1 also is stable with the cation bound within a pocket created by the distal portion of the S-loop and the interim loop connecting the Cat and Fib domains. This stability is also confirmed by the low RMSD and RMSF values for the Cat domain and those of Zn^2+^ ion 1 and Ca^2+^ ion 1 with both ions staying closely associated with the Cat domain of the protein and approximate to their crystallographically defined positions during the simulation (Table 2 and Table 3 and Figure 3 and Figure 4). The other associated metal ions of the Cat domain do not share this degree of stability (Table 3). Zn^2+^ ion 2 remains associated with the S-loop, but loses contact with the crystallographically demonstrated interactions with the His residues on the β4- and β5-sheets. This may not be unexpected, since conformational flexibility within the S-loop region, as was reported previously [29] and may allow for changes in the binding pocket conformation necessary for substrate recognition. Ca^2+^ ion 2 does not remain in close contact with the β1-β3-loop and diffuses out of the binding pocket (Figure 3 and Figure 4). The Hpx associated Ca^2+^ ion 3 remains close to its crystallographically observed binding site (Figure 3 and Figure 4) with a relatively low RMSF (Table 3), but to a lower degree that either Zn^2+^ ion 1 or Ca^2+^ ion 1.

The majority of the conformational fluctuations within MMP-2 are within the Fib domain (Figure 3 and Figure 4 and Table 2). The three type II Fib subdomains are highly flexible, partly due to the large amount of β-turn/bend and coil structure within the subdomains. This degree of flexibility may be important for interactions between MMP-2 and its collagen substrates [25,26,27,28]. There is also clearly an inter-domain interaction that occurs between Hpx and the Cat and Fib domains mediated by the Lnk region. The Lnk region acts as a complex hinge allowing the COM distance between the Cat/Fib and Hpx domains to open and close while changing the orientation of the Hpx domain from an edge view to an end on view (Figure 3). This rotation of the Hpx domain in relation to the Cat domain has also been noted in prior simulations [14,15].

The radial distribution functions for the Zn^2+^ ions demonstrated peaks at 0.20 nm and 0.42 nm with troughs located at 0.31 nm and 0.50 nm. The Ca^2+^ ions have slightly enlarged the first and second hydration shells, with peaks at 0.25 and 0.44 nm, and troughs at 0.31 nm and 0.55 nm, respectively. The statistical densities of the first and second hydration shells (ρ (1st shell) and ρ (2nd shell)) are given in Table 4. These values represent the probabilities of a water molecule(s) occupying the volume of the hydration shell at any sampled time during the trajectory. There is only a minor difference in the 1st hydration shell comparing Zn^2+^ ion 1 and Zn^2+^ ion 2 with more significant changes in the 2nd hydration shell. This is consistent with Zn^2+^ ion 1 sequestered within the catalytic pocket of the enzyme while Zn^2+^ ion 2 dissociates from His^178^, His^193^, and His^206^ and interacting with Asp^180^ on the flexible S-loop. These results are also consistent with the solvent accessible surface area (SASA) which is much lower for Zn^2+^ ion 1 than Zn^2+^ ion 2. The results for the Ca^2+^ ions demonstrate that Ca^2+^ ion 1 is solvent sequestered with a low probability of water within the first hydration shell. Ca^2+^ ion 2 which dissociates from its binding site early in the simulation and appears to diffuse freely is similar the Ca^2+^ ion3 which stays in closer proximity to the Hpx domain, but does not stay in close contact with its crystallographically associated residues. The SASA values in Table 4 also reflect this. The coordination numbers (CN) for the ions are given below. The value of 1 to 1.5 for Zn^2+^ ion 1 is consistent with prior studies [30,31]. The CN was also be calculated for Glu^404^ and is equal 1, in agreement with this residue’s catalytic importance and its proximity to the catalytic Zn^2+^ ion 1. The sequestered nature of Ca^2+^ ion 1 indicates only one associated water molecule while Ca^2+^ ion 2 and Ca^2+^ ion 3 are more normally hydrated.

### 2.3. Domain Movement Analysis

DCCM (Figure 5) demonstrated that motions within Cat, Fib and Hpx domains and Lnk region are, for the most part, highly correlated with a few exceptions. The first five principle components (PC) account for a total of 93.7% of the total domain motions (Figure 6). The plots of the Cα-trace RMSF of the projection of the trajectory onto the first five eigenvectors are shown (Appendix A). For PC1, the majority of the contributions arises from the first two subdomains of Fib, the Hpx domains and Lnk region. PC2, has significant motions in the third subdomain of Fib and Hpx. There are also anti-correlated motions with the Cat domain involving the α1-helix and β1- through β5-sheets that contain the active site. Motions along PC3 through PC5 represent minor fluctuations within the domains and global conformation. The distal portion of the Cat domain which contains the active site has correlated motions with the first and third subdomains of the Fib domain and the first and second blades of the Hpx domain. There are also anti-correlated motions between the active site on the Cat domain and the second subdomain of the Fib domain and the third and fourth blades of the Hpx domain.

### 2.4. Protein-Metal Ion Interaction Energies

MMP-2 has a high affinity for the bound divalent cations (Table 5) with a major contribution from the electrostatic interactions. The solvation term ∆G_polar_ is unfavorable particularly for the catalytic Zn^2+^ ion 1, the structural Ca^2+^ ion 1, and Ca^2+^ ion 3 and the ∆G_non-polar_ is only weakly favorable. For the weakly bound Zn^2+^ ion 2 and the freely diffusing Ca^2+^ ion 2, the ∆G_polar_ is significantly smaller. The entropic contributions to MMP-2 divalent cation interaction energies are very small, but positive-indicating decreased system entropy with the binding of the metal ions to the peptide. The entropic term is lowest for those ions that for those metal ions (Ca^2+^ ion 2 and Ca^2+^ ion 3) that either diffuse freely away from their crystallographically determined binding sites or do not form close associations and stable binding geometries with the electronegative backbone and sidechain atoms of MMP-2 [32].

Residue contributions to the binding energy with their associated interatomic distances and geometries are given in Table 6, only those protein residues within the first coordination sphere of the divalent metal ion are noted with the exceptions of Ca^2+^ ion 2 and Ca^2+^ ion 3 which are discussed below [17,18]. The catalytic Zn^2+^ ion 1 maintains interactions with His^403^, His^407^, and His^413^, similar to what is observed in the x-ray crystal structure. The Glu^404^ sidechain Oε atoms are in closer proximity than what is observed in the crystal structure; however, with distances that are consistent with prior simulations [30,31]. The binding geometry for the bound His residues is trigonal bipyramidal; however, if coordinated hydration water is considered, this geometry would be tetrahedral. Other important interactions are also noted between Zn^2+^ ion 1 and Asp and Glu residues within the Cat domain. These residues contribute significantly to the MMP-2-Zn^2+^ ion 1 interaction energy despite being outside what is considered to be the normal coordination sphere of the ion. This is not unexpected, given the strong long-distance coulombic interactions involved (Equation (2)). Zn^2+^ ion 2 loses contact with His^178^, His^193^, and His^206^ shifting to a more linear coordination geometry that is depended on a strong interaction with the Oδ atoms of Asp^180^. Comparison to prior simulation studies is not possible for the Zn^2+^ ions, since the parameters used represented the interactions between the divalent metal cation and the x-ray crystal structure associated residue sidechains as harmonic potentials [30]. This is an important point, since prior investigators have identified variations in the stoichiometry of the MMP-2-Zn^2+^ interaction that are strongly dependent on the purification procedure used possibly indicating that Zn ^2+^ ion 2 may be able to dissociate from its binding site [33]. It should be noted that the current study assumes that all His residue sidechains are uncharged with a pK_a_ = 6.0 at a system pH = 7.0 [34]. This assumption cannot account for local environments that may cause the His sidechains to be protonated effecting Zn^2+^ binding as has been demonstrated for His residue interactions with other divalent cations and their metal binding site [35].

Ca^2+^ ion 1 has strong interactions with the adjacent Oδ atoms of Asp and Oε atoms of Glu of adjacent residues. There is a shift from the divalent cation to backbone carbonyl oxygen interactions that are observed in the crystal structure to interactions dominated by the acidic sidechain groups. The coordination geometry changes from pentagonal pyramidal to a square planar geometry. The strong interactions and coordination with these sidechains is expected and has been previously observed for other systems [36]. The results for Ca^2+^ ion 1 are the same as noted in prior simulations with a shift from backbone carbonyl interactions, which may represent crystal packing forces to interactions with the carboxyl sidechains [30]. In general, the favorability of the interaction is Glu>Asp, with the difference attributed to the increased flexibility of the Glu residue, secondary to the presence of the extra methylene group. The binding of this ion is also similar to that of the catalytic Zn^2+^ ion 1 in that adjacent, but non-coordinated electronegative Asp and Glu residues make significant contributions to its binding energy secondary to the strong long range coulomb interactions. Ca^2+^ ion 2 and Ca^2+^ ion 3 represent special cases. Ca^2+^ ion 2 freely diffuses out of its binding site, and any strong interactions with electrostatic sidechains are transient. The x-ray crystal structure associated atoms are given to illustrate this, Although prior simulations do note the Ca^2+^ ion 2 leaving its binding site, they do note that the associated interactions are weak and the study is most likely limited by the short simulation time (10 ns) [30]. Ca^2+^ ion 3 is more stable in its RMSF compared to Ca^2+^ ion 2 (Table 3); however, it is still much more variable than the other associated ions. This ion breaks its association with the backbone carbonyl atoms of Asp^476^, Asp ^521^, Asp^569^ and Asp^618^ to form strong interactions with multiple Asp carboxyl oxygens as given in Table 6. This result is most likely due to the change from crystal packing forces to that of an aqueous environment [36].

## 3. Materials and Methods

### 3.1. Matrix Metalloprotease-2 Starting Conformation

Initial coordinates were obtained from the X-ray crystal structure of the Glu^404^ to Ala^404^ mutant of the human MMP-2 (PDB ID: 1CK7) [13]. Crystallographically resolved water and sulfate ions were removed while the protein bound Zn^2+^, Ca^2+^, Na^+^, and Cl^−^ ions where retained. Residues 31–109 were removed as in the biologically active form of the enzyme. The crystallography non-resolved loop from residues Asp^450^-Thr^460^ was build using the homology modelling script of YASARA [16]. The coordinates for the sidechain of Glu^404^ and the Zn^2+^ coordinated water molecule that replaces Cys^102^ at the enzyme active site were derived from the MMP-13 crystal structure (PDB ID: 1XUD) by the least squares fitting of the backbone and Zn^2+^ atoms of both x-ray structures [37]. Sidechain protonation states of the Zn^2+^ associated His residues were assigned based on the 1CK7 crystal structure as follows: HND1 for His^178^, His^403^, His^193^, His^407^, His^413^, and HNE2 for His^206^. The remaining Histidine residues (His^163^, His^276^, and His^628^) were assigned automatically with HNE2 atoms using the *pdb2gmx* module of GROMACS version 5.1.2 [38,39]. The protonation state and charges of all other residues within the protein were set to correspond to a pH of 7.0 with the His residues sidechains assigned in their uncharged state consistent with NMR titration studies [34].

### 3.2. Molecular Dynamics

Simulations were performed using the CHARMM36m force field with modified TIP3Pm water model and the CM model of divalent metal cation parameters of Li et al. as implemented in GROMACS version 5.1.2 [Zn^2+^ (σ = 0.226466454151 nm, ε = 0.01381916624 kJ mol^−1^) and Ca^2+^ (σ = 0.293818397243 nm, ε = 0.44320568080 kJ mol^−1^)] [40,41,42,43,44,45]. The CM model attempts to balance hydration free energy by optimizing the ion-oxygen distance in the first solvation shell. The metal ions are represented with a standard Lennard-Jones and coulomb potential energy functions [46]:(2)$$E_{ion}=\overset{N}{\underset{j}{\sum }}4\varepsilon_{ij}\left[{\left(\frac{\sigma_{ij}}{r_{ij}}\right)}^{12}-{\left(\frac{\sigma_{ij}}{r_{ij}}\right)}^{6}\right]+\overset{N}{\underset{j}{\sum }}\frac{q_{i}q_{j}}{4\pi \varepsilon_{0}r_{ij}},$$
where, σ_ij_, is the distance between two atoms at their lowest potential energy calculated as an arithmetic mean,
(3)$$\sigma_{ij}=\frac{\left(\sigma_{i}+\sigma_{j}\right)}{2},$$
ε_ij_, is the depth of the potential energy well calculated as a geometric mean,
(4)$$\varepsilon_{ij}=\sqrt[{2}]{\left(\varepsilon_{i}•\varepsilon_{j}\right)}$$
q_i_ and q_j_ are the respective point charges, r_ij_ is the distance separating the two atoms, ε_0_ is the dielectric constant, and *i* and *j* are atom indices. The system was solvated in a truncated dodecahedron with 45105 TIP3Pm water molecules. The minimal distance of the protein to the edge of the dodecahedron was 1.4 nm. The system was neutralized with 141 and 132, Na^+^ and Cl^−^ ions, respectively, so that the final concentration of the NaCl was set to 150 mM; the initially retained protein bound Zn^2+^, Ca^2+^, Na^+^, and Cl^−^ ions are not included. The size of the box was 1458.26 nm^3^, containing 8487 protein and divalent metal cation atoms and 45106 water molecules giving an MMP-2 concentration of 1.1 mM, a Zn^2+^ concentration of 2.2 mM and a Ca^2+^ concentration of 3.3 mM. The mean ± standard deviation concentrations of non-protein bound Zn^2+^ and Ca^2+^ in the plasma of healthy human adults are 81 ± 16 nM and 1.19 ± 0.06 mM, respectively [47,48]. The system was subjected to 5000 steps of steepest descent energy minimization, allowing all bond distances and angles to relax. This was followed with 10 ns of NVT simulation at 310 K so that the position of the protein heavy atoms and retained Zn^2+^, Ca^2+^, Na^+^, and Cl^−^ ions and catalytic Zn^2+^ associated water were constrained to their energy-minimized coordinates with force constant of 1000 kJ mol^−1^. The solvent and non-restrained Na^+^ and Cl^−^ ions were then subjected to 10 ns of NPT simulation at 310 K and 101.325 kPa using Berendsen temperature and pressure scaling with a relaxation constant of 0.1 ps and 4.5 × 10^−5^ bar^−1^ isothermal compressibility [49]. The heavy atom restraints were removed and the system subjected to 10 ns of NPT dynamics (310 K and 101.325 kPa). Velocities were assigned and rescaled stochastically and the pressure coupled with the Berendsen method (relaxation constant, 0.1 ps and isothermal compressibility, 4.5 × 10^−5^ bar^−1^) [50]. The production run consisted of 1.0 µs NPT simulation (310 K and 101.325 kPa). Trajectories were integrated with a 2 fs time step. All bonds were constrained to their correct length using LINCS, with a warning angle of 30° [51,52]. The long-range electrostatic interactions were calculated using the PME method with 1.2 nm cutoff distance and 0.15 nm Fourier spacing [53]. Van der Waals interactions were calculated using short-range and long-range cutoffs of 1.0 and 1.2 nm respectively with a linear smoothing function. The temperature was maintained by the stochastic velocity-rescaling method, and the system was coupled to a Parrinello-Rahman barostat [50,54].

### 3.3. Biophysical Properties

The Cα-trace root mean square deviation (RMSD) and per-residue root mean square fluctuation (RMSF) from the average sampled peptide conformation were calculated with the *g_rmsd* and *g_rmsf* utilities of GROMACS, respectively [39]. The >50% of the simulation time sampled per-residue DSSP assigned [24] secondary structure (α-helix, β-sheet, β-bend/turn, and coil) were determined using the *do_dssp* utility of GROMACS and an in house *perl* script to calculate sampling statistics. The hydration of the Zn^2+^ and Ca^2+^ was determined by calculating the radial distribution function for the oxygen atom of the surrounding water molecules using the *g_rdf* utility in GROMACS [39]. The distances from the ion to the first and second hydration shells is determined by examining the peaks and troughs of the associated RDF. The integral of the RDF (g(r)):(5)$$\rho_{r}=\int_{0}^{r}g\left(r\right)dr,$$
is the statistical density of the surrounding solvent and represents the probability of a solvent molecule being located within r distance of the central atom or group at any sampled time. At an infinite distance this probability is by definition 1.0. The coordination number (CN) can then be calculated as:(6)$$CN=4\cdot \pi \cdot \rho \int_{0}^{r}g\left(r\right)r^{2}dr,$$
where ρ is the density and r the distance from the ion [55,56]. 

### 3.4. Conformational Analysis

The time-dependent ϕ/ψ dihedral angles from residues 111 to 659 were calculated using the *g_rama* utility of GROMACS and transformed into appropriate input files using a python script. The dihedral principle component analysis (dPCA) program was provided by Dr. Yuguang Mu [57,58]. Lowest energy conformations were identified by projecting the trajectories of the first two principal components (dPC1 and dPC2) onto a two-dimensional free energy (∆G) landscape:(7)$$\Delta G=-R\cdot T\cdot ln\frac{\rho_{x,y}}{\rho_{max}},$$
where R is the universal gas constant, T is the temperature, x and y are the first two dihedral principal components from the trajectory. The free energy (∆G) landscape was calculated by dividing the dPC1-dPC2 subspace into grids creating a 2D histogram of the sampled phase space and calculating the probability ρ_x,y_ where ρ_max_ is the grid value corresponding to the maximum probability of occurrence. The free energy landscape was visualized using the *scatterplot3D, akima*, and *latticeExtra* packages in the R [59,60,61]. K-means clustering (*cluster* package in R) was used to identify families of conformations and the lowest energy conformations extracted for analysis [22,23]. The optimal clustering was determined using a combination of visual inspection, sum of squared error (SSE), average silhouette width (S_AVG_), silhouette coefficient (SC) and distribution plots [62,63,64,65]. Conformations and secondary structural elements were rendered using YASARA [16].

### 3.5. Dynamic Cross-Correlation Matrix

Domains movement and correlations were evaluated by dynamic cross-correlation matrices (DCCM) calculated from the Cα-trace covariance matrix principal components using the *GeoStaS* method (*Bio3D* package in R) [66,67,68]. Results are displayed as a color coded matrix of Pearson correlation coefficients with a value of −1 indicated completely anti-correlated motions and a value of +1 indicating completely correlated motions [69,70].

### 3.6. Interaction Energy

The free energy of binding between the metal cations and protein was calculated using the *g_mmpbsa* program [71]. The polar component of the solvation energy was calculated using the Poisson-Boltzmann equation and non-polar component calculated from the solvent-accessible surface area approximation [72,73]. Dielectric constants for the solute and water were 4 and 80, respectively. The entropic contribution to the binding energy was determined using the interaction entropy method of Zhang and coworkers [32,74]. The method defines the entropic contribution to the free energy term as:(8)$$-T\Delta S=R\cdot T\cdot ln\langle e^{\beta \Delta E_{pl}^{int}}\rangle ,$$
where R is the universal gas constant, T the temperature, β is the thermodynamics beta:(9)$$\beta =\;\frac{1}{k_{B}T}\;,$$
β is Boltzmann’s constant, T is temperature, and ∆E_pl_^int^ is defined as:(10)$$\Delta E_{pl}^{int}=E_{pl}^{int}-\langle E_{pl}^{int}\rangle ,$$
$\langle E_{pl}^{int}\rangle$ is the ensemble averaged protein-ligand interaction energy, and E_pl_^int^ is the protein-ligand interaction energy for a sampled conformation. For molecular dynamics simulations containing N sampled conformations, the interaction entropy can then be calculated as:(11)$$\langle e^{\beta \Delta E_{pl}^{int}}\rangle =\frac{1}{N}\sum e^{\beta \Delta E_{pl}^{int}}$$

The trajectory was sampled every 0.1 ns for the equilibrium phase (200 ns to 1000 ns). A bootstrap analysis (n = 5000) was performed to obtain standard errors, and the residue contributions to the binding energy were also calculated.

The residue contributions to binding were deconvoluted. To determine the most significant residue interactions between MMP-2 and the divalent cations, an outlier analysis was performed to identify statistically significant interactions. The distribution of interaction energies was not Gaussian (normal). In the setting of non-normal distributions, the method of Tukey’s fences can be used to identify those observations that are outside of the expected fluctuations within the data [75]. Tukey’s fences defines the minimum and maximum values of the interaction energy:(12)[MinimumValue,MaximumValue],
such that measurements less than or equal to the minimum value or greater than or equal to the maximum value are considered statistical outliers. The respective minimum and maximum values of the fences, are defined as:(13)$$\left[Q_{1}-k\left(Q_{3}-Q_{1}\right),Q_{3}+k\left(Q_{3}-Q_{1}\right)\right],$$
where Q_1_, Q_3_ are the interquartile values, and k is the constant that defines the outlier range (k = 1.5 is an outlier, k = 3.0 is an extreme outlier) [75].

## 4. Conclusions

We report a microsecond scale molecular dynamics analysis of the full biologically active protein (Cat, Fib, and Hpx domains with Lnk region) with its crystallographically associated (structural) divalent metal ions (Zn^2+^ and Ca^2+^). Our model indicates that MMP-2 undergoes significant inter domain motions. The R_g_ and COM data demonstrate five macro distributions of conformations based on size. The dPCA data is consistent with ten families of conformations. The three lowest energy populations of conformations differ predominantly in the orientation of the Hpx domain in relation to the Cat and Fib domains. This is confirmed by the DCCM analysis, where the difference in orientation of the Hpx to the Cat and Fib domains comprises the first two principle components. These inter domain movements are facilitated by the flexible linker region Gly^446^ through Asp^476^ and may play an important role in collagen substrate binding, unravelling and subsequent catalysis.

Zn^2+^ ion 1 (catalytic ion) and Ca^2+^ ion 1 are tightly bound within their crystallographically defined pockets. Zn^2+^ ion 2 was more flexible and associated with the S-loop which has demonstrated increased flexibility in both prior simulations and our own; however, the use of short simulation time (10 ns) and a harmonic potential representing the interaction between the Zn^2+^ ion and the associated His and Asp/Glu residues may have artificially stabilized the protein-Zn^2+^ ion 2 interaction. Ca^2+^ ion 2 and Ca^2+^ ion 3 do not remain in their crystallographically defined positions. Although Ca^2+^ ion 3 is more closely associated with the Hpx domain, the carboxyl sidechains of the Asp residues that contribute to its interaction are at distances well outside of the coordination sphere.

The current non-bonded model of Li et al. with the CHARMM36m force field appears to be a reasonable model of protein metal cation interactions in MMP-2 [45]. The data suggest that further models should retain Zn^2+^ ion 1, Zn^2+^ ion 2 and Ca^2+^ ion 1, since they play important chemical and conformational roles and remove Ca^2+^ ion 2 and possibly Ca^2+^ ion 3, since their role is minimal. Protein-ligand binding studies should also include multiple conformations to account for the mobility of the Hpx domain with respect to the Cat and Fib domains. We do, however, acknowledge that the Li et al. with the CHARMM36m force field has known limitations. Accurate protein-metal ion interaction energies and coordination geometries require either the use of a carefully parameterized polarizable force field or more computationally expensive QM/MM simulations.

## Figures and Tables

**Figure 1 ijms-20-04194-f001:**
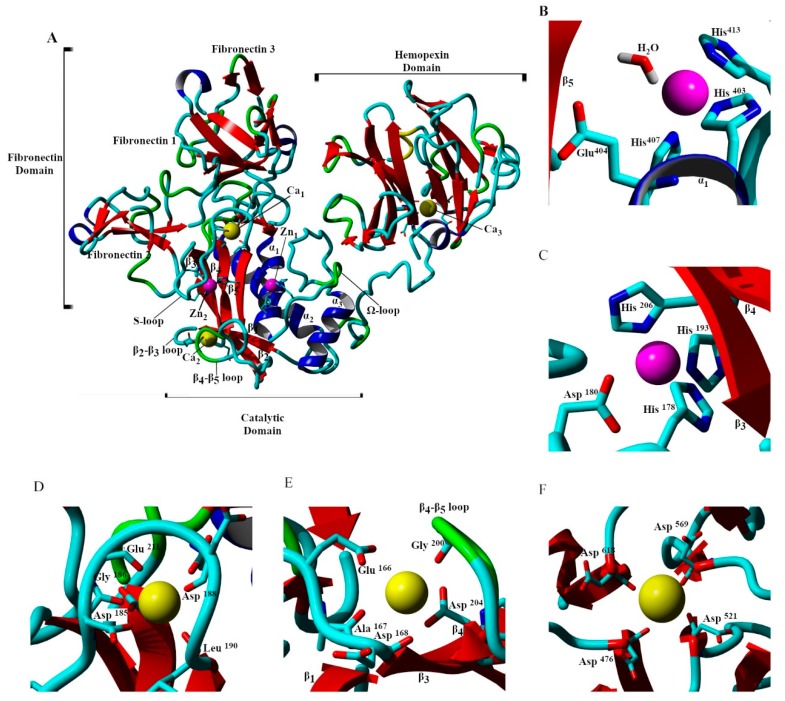
The ribbon diagram of the X-ray crystal structure of 1CK7. The pro-peptide (Pro^31^-Gln^109^) region is removed, and the unresolved link (Asp^450^-Thr^460^) connecting the Cat and Hpx domains built with YASARA [16]. Domains, subdomains, and secondary structural features of the catalytic domain are labeled accordingly (blue, α-helix; red, β-sheet; green, β-turn/bend; and aqua, coil) (**A**). The associated ions are shown as van der Waals radii with Zn^2+^ pink and Ca^2+^ yellow. The catalytic Zn^2+^ ion 1 is bound to His^403^, Glu^404^, His^407^, His^413^, and catalytic water (**B**). Zn^2+^ ion 2 is bound to His^178^, Asp^180^, His^193^, and His^206^ (**C**). Ca^2+^ ion 1 is bound to Asp^185^, Gly^186^, Asp^188^, Leu^198^, Asp^208^, and Glu^211^ (**D**). Ca^2+^ ion 2 is bound to Glu^166^, Ala^167^, Asp^168^, Gly^200^, Gly^202^, and Asp^204^ (**E**). Ca ^2+^ ion 3 is bound to Asp ^476^, Asp^521^, Asp^569^, and Asp^618^ (**F**).

**Figure 2 ijms-20-04194-f002:**
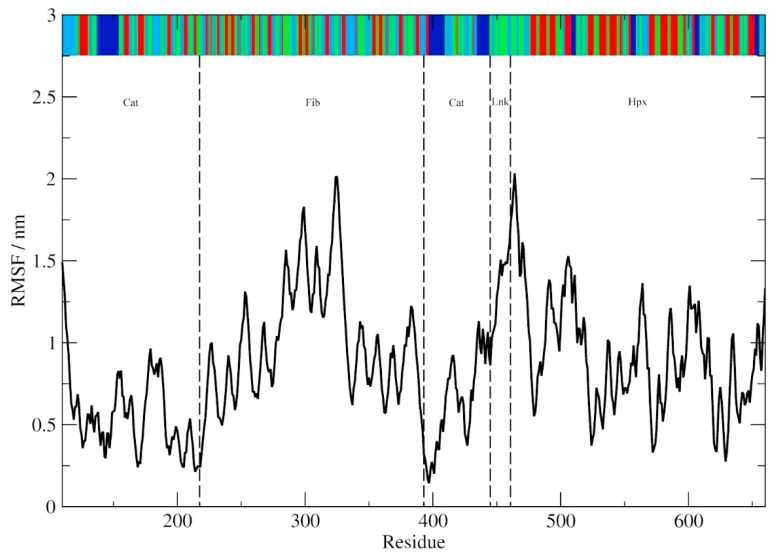
The Cα-trace root mean square fluctuation (RMSF). The DSSP assigned secondary structure (sampled >50% of the simulation time) is shown at the top of the graph (blue, α-helix; red, β-sheet; green, β-turn/bend; and aqua, coil). The Cat, Fib and Hpx domains and Lnk regions are demarcated with dotted lines.

**Figure 3 ijms-20-04194-f003:**
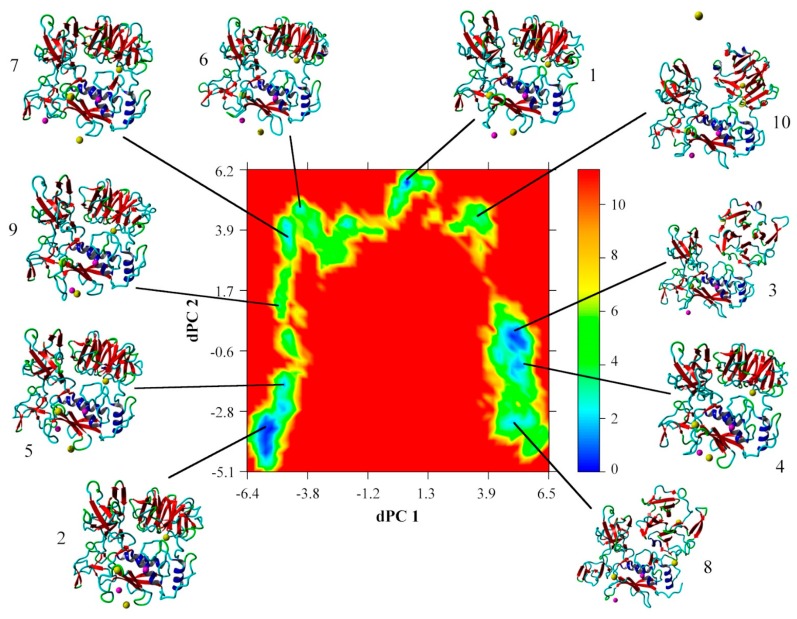
Free energy landscape (kJ mol^−1^) as a function of the first two dihedral principle components (dPC1 and dPC2); the lowest energy conformations of each family as identified by k-means clustering are shown. Secondary structural motifs and ions are shown (blue, α-helix; red, β-sheet; green, β-turn/bend; aqua, coil; pink, Zn^2+^; and yellow, Ca^2+^).

**Figure 4 ijms-20-04194-f004:**
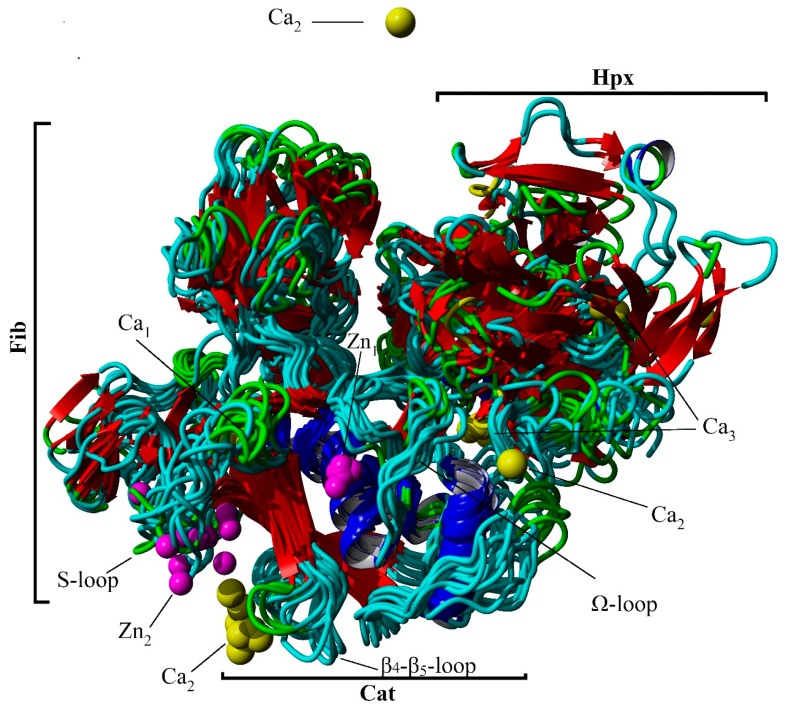
Cα-trace overlay of 10 cluster centroid structures from the cluster analysis of dPC1 and dPC2. Secondary structural motifs and ions are shown (blue, α-helix; red, β-sheet; green, β-turn/bend; aqua, coil; pink, Zn^2+^; and yellow, Ca^2+^).

**Figure 5 ijms-20-04194-f005:**
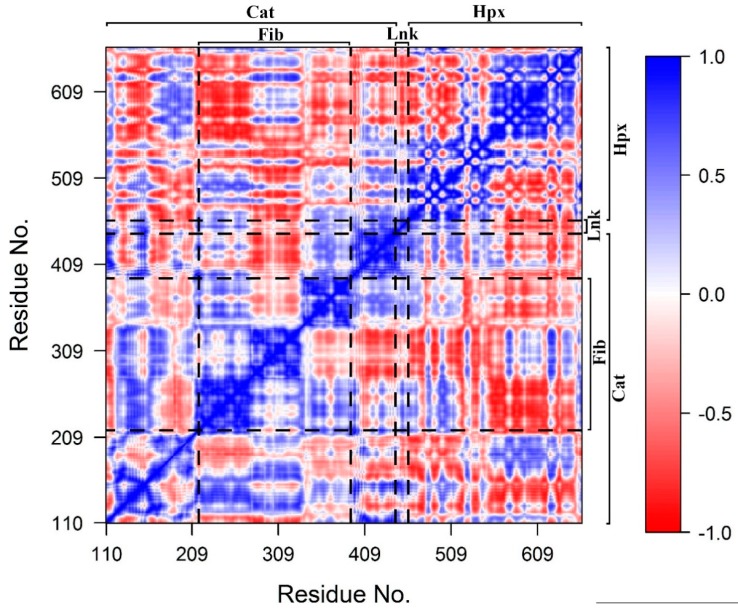
Dynamic cross-correlation matrix. Values range from -1 (complete anti-correlation) to +1 (complete correlation). The Cat, Fib, and Hpx domains and Lnk regions are demarcated with dashed lines.

**Figure 6 ijms-20-04194-f006:**
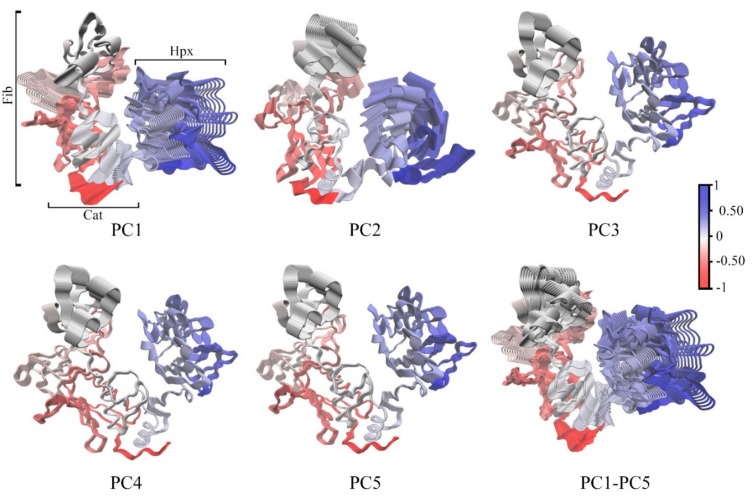
Cα-trace of the first five principle components (PC1 through PC5) and an overlay of all principle components PC1-PC5. Correlated domain movements are indicated in blue and anti-correlated are in red.

**Table 1 ijms-20-04194-t001:** Distances between ions and the interacting matrix metaloproteinase-2 (MMP-2) residue atoms and associated binding geometry identified from the 1CK7 X-ray crystal structure. Binding geometries and distances were determined using *CheckMyMetal*
^a,b,c^.

Protein Atom	Geometry	Distance/nm				
		Zn^2+^ ion 1	Zn^2+^ ion 2	Ca^2+^ ion 1	Ca^2+^ ion 2	Ca^2+^ ion 3
His^403^:Nε2	tetrahedral	0.23				
His^407^:Nε2	tetrahedral	0.22				
His^413^:Nε2	tetrahedral	0.25				
Water:O	tetrahedral	0.25				
Glu^404^:Oε1		0.83				
Glu^404^:Oε2		0.76				
His^178^:Nε2	trigonal bipyramidal		0.21			
Asp^180^:Oδ2	trigonal bipyramidal		0.23			
His^193^:Nε2	trigonal bipyramidal		0.21			
His^206^:Nδ1	trigonal bipyramidal		0.21			
Asp^185^:C=O	octahedral			0.29		
Gly^186^:C=O	octahedral			0.24		
Asp^188^:C=O	octahedral			0.26		
Leu^198^:C=O	octahedral			0.25		
Asp^208^:Oδ2	octahedral			0.25		
Glu^211^:Oε2	octahedral			0.27		
Ala^167^:C=0	poorly coordinated				0.29	
Asp^168^:C=O	poorly coordinated				0.29	
Gly^200^:C=O	poorly coordinated				0.27	
Asp^476^:C=O	square planar					0.25
Asp^521^:C=O	square planar					0.28
Asp^569^:C=O	square planar					0.27
Asp^618^:C=O	square planar					0.27

^a^ Zn^2+^ ion 1 is the catalytic ion; ^b^ Glu^404^ is critical to catalytic activity; ^c^ Location and orientation of the catalytic water was derived from the sidechain of Cys^102^ and the X-ray crystal structure of MMP-13 (PDB (protein databank) ID: 1XUD).

**Table 2 ijms-20-04194-t002:** Average root mean square deviation (RMSD) of the Cα-trace of MMP-2 as a whole and divided into its individual domains: Cat, Fib, and Hpx with the Lnk region ^a,b^.

	All	Cat w/ Fib	Fib	Cat w/o Fib	Hpx	Lnk
RMSD/nm	3.20 ± 0.14	3.34 ± 0.02	3.69 ± 0.03	0.50 ± 0.03	0.34 ± 0.04	0.54 ± 0.08

^a^ w/ is with; ^b^ w/o is without.

**Table 3 ijms-20-04194-t003:** Average RMSF of the associated divalent cations to the Cα-trace of MMP-2 ^a^.

	Zn^2+^ ion 1	Zn^2+^ ion 2	Ca^2+^ ion 1	Ca^2+^ ion 2	Ca^2+^ ion 3
RMSF/nm	0.593	0.882	0.571	2.906	1.062

^a^ Zn^2+^ ion 1 is the catalytic ion.

**Table 4 ijms-20-04194-t004:** Solvation properties of the associated divalent cations. SASA, solvent exposed surface area.

Property	Zn^2+^ ion 1	Zn^2+^ ion 2	Ca^2+^ ion 1	Ca^2+^ ion 2	Ca^2+^ ion 3
**ρ (1st shell)**	0.20	0.26	0.07	0.23	0.25
**ρ (2nd shell)**	0.29	0.41	0.22	0.44	0.41
**SASA/nm^2^**	0.002 ± 0.010	0.046 ± 0.030	0.021 ± 0.013	0.311 ± 0.110	0.302 ± 0.119
**CN**	1 to 2	2	1	3	3

^a^ ρ is the statistical density of the hydration shell; ^b^ SASA is the solvent accessible surface area; ^c^ CN is the coordination number.

**Table 5 ijms-20-04194-t005:** Binding energies between MMP-2 and the associated divalent cations as determined by the MMPBSA-IE and interaction entropy methods with its associated components.

Mean ± Standard Deviation/kJ mol^−1^							
Ion(s)	∆E_vdw_	∆E_elec_	∆G_polar_	∆G_non-polar_	−T∆S	∆E_binding_	∆G_binding_
All ions	95.04 ± 2.92	−9264.45 ± 194.34	3142.58 ± 81.6	−3.95 ± 0.13	6.77	−6020.73 ± 116.75	−6013.96 ± 116.75
Zn^2+^ ion 1	25.41 ± 0.77	−2427.66 ± 72.19	1291.3 ± 42.75	−0.88 ± 0.04	0.63	−1114.16 ± 27.12	−1113.51 ± 27.12
Zn^2+^ ion 2	13.79 ± 0.96	−799.35 ± 63.45	188.73 ± 35.14	−0.47 ± 0.04	0.52	−598.61 ± 29.09	−598.09 ± 29.09
Ca^2+^ ion 1	38.15 ± 1.90	−1796.42 ± 89.21	605.84 ± 30.55	−1.14 ± 0.07	1.03	−1154.73 ± 56.65	−1153.69 ± 56.65
Ca^2+^ ion 2	6.52 ± 0.58	−367.21 ± 36.96	73.71 ± 9.94	−0.35 ± 0.05	0.18	−289.24 ± 27.33	−298.07 ± 27.33
Ca^2+^ ion 3	14.94 ± 0.57	−1480.59 ± 42.61	320.38 ± 11.31	−1.32 ± 0.05	0.23	−1147.96 ± 30.78	−1147.73 ± 30.78

**Table 6 ijms-20-04194-t006:** Distances of ions to the interacting protein residue atoms identified from the 1CK7 X-ray crystal. Statistically significant protein residue atoms as identified by outlier analysis with the associated per residue interaction energies and binding geometry. 1CK7 identified interactions are marked with a dagger (†); statistically significant interactions are marked with an asterisk (*); and binding geometries were determined for those atoms within 0.35 nm. Distance is given in nm and energies in kJ mol^−1 a,b,c^.

Protein Atom	∆E_binding_	Geometry	Zn^2+^ Ion 1	Zn^2+^ ion 2	Ca^2+^ ion 1	Ca^2+^ ion 2	Ca^2+^ ion 3
†*Glu^404^:Oε1	−154.8223		0.46 ± 0.05				
†*Glu^404^:Oε2			0.48 ± 0.06				
†*His^403^:Nε2	−72.1511	trigonal pyramidal	0.21 ± 0.01				
†*His^407^:Nε2	−66.3450	trigonal pyramidal	0.21 ± 0.01				
†*His^413^:Nε2	−61.5998	trigonal pyramidal	0.21 ± 0.01				
†*Asp^180^:Oδ1	−92.3424	linear		0.19 ± 0.01			
†*Asp^180^:Oδ2		linear		0.19 ± 0.01			
†*Asp^185^:Oδ1		seesaw			0.29 ± 0.16		
†*Asp^185^:Oδ2		seesaw			0.28 ± 0.16		
†*Glu^211^:Oε1	−147.6924	seesaw			0.33 ± 0.11		
†*Glu^211^:Oε2		seesaw			0.35 ± 0.10		
†*Asp^208^:Oδ1	−142.7074	seesaw			0.34 ± 0.09		
†*Asp^208^:Oδ2		seesaw			0.24 ± 0.02		
*Asp^210^:Oδ1	−112.8532	seesaw			0.33 ± 0.14		
*Asp^210^:Oδ2		seesaw			0.35 ± 0.14		
Asp^168^:Oδ1		N/C				2.57 ± 1.59	
Asp^168^:Oδ2		N/C				2.54 ± 1.60	
†Ala^167^:C=O	−1.3321	N/C				2.52 ± 1.75	
†Gly^200^:C=O	0.3799	N/C				2.57 ± 1.70	
†Gly^202^:C=O	−1.0605	N/C				2.55 ± 1.56	
*Asp^521^:Oδ2		N/C					0.98 ± 0.16
†*Asp^569^:C=O	−79.0260	N/C					0.71 ± 0.18
*Asp^569^:Oδ1		N/C					0.92 ± 0.15
*Asp^569^:Oδ2		N/C					0.91 ± 0.15
*Asp^490^:Oδ1	−69.0106	N/C					1.19 ± 0.11
*Asp^490^:Oδ2		N/C					1.19 ± 0.11
*Asp^615^:Oδ1	−68.1150	N/C					1.29 ± 0.14
*Asp^615^:Oδ2		N/C					1.30 ± 0.14
*Asp^153^:Oδ1	−63.3125	N/C					2.52 ± 0.96
*Asp^153^:Oδ2		N/C					2.51 ± 0.95
*Asp^472^:Oδ1	−54.3377	N/C					1.54 ± 0.15
*Asp^472^:Oδ2		N/C					1.54 ± 0.15

^a^ Zn^2+^ ion 1 is the catalytic ion; ^b^ Glu^404^ is critical to catalytic activity; ^C^ N/C not coordinated.

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
