# Peer review of "Conformation and Domain Movement Analysis of Human Matrix Metalloproteinase-2: Role of Associated Zn2+ and Ca2+ Ions"

_ijms, 2019, doi:10.3390/ijms20174194_

Round 1

Reviewer 1 Report

In this paper, the authors describe a conformational analysis of a 1 microsecond long molecular dynamics simulation of the Human Matrix Metalloproteinase-2. They describe the relative movements of the 3 domains of the protein, as well as the stability of the divalent cations observed in the X-ray structure.  

My main concern with this paper is that the objectives of the work are not really clearly defined. The authors used many tools to analyze their MD simulation. But it is hard to understand what they really want to show with these analyses and the relevance of their findings in a biological context, and compare them with available literature. For example, the authors cite three simulation papers on the same protein, but the comparison of their results with them is minimal. I think the authors should deepen their analysis to fully convince the reader of the relevance of their work. That is why I think the paper might be accepted for publication but only after major revision.

I list below some specific points that I think the authors should consider to improve the paper:

- there are long tables with lot of numbers (table 1 and table 6 especially). If the reader is not a specialist of the system, I guess he will not be able to decide what to look at, and the authors do not really help them in the text. Are these data really necessary in the main text? If so, I think the authors should spend more time in the text explaining which numbers are relevant and tell why. If not, maybe the tables could be shortened conserving only the most important numbers and putting all the other data as supplementary information.

- lines 88-92: the authors claim that the MD simulation has converged based on three quantities: configuration entropy, radius of gyration, and center of mass distance between domains. Obviously, looking at the two last measure, the simulation is not converged. A large variation is seen during the last 200ns. Maybe this is only a temporary conformational change, but it would be necessary to continue the simulations to establish it.

- line 83: I don’t know what “interaction entropy binding energies” means. I think the authors meant “binding free energies” that are actually computed with MM-PBSA.

- In table 2, authors give some numbers about RMSD of individual domains of the molecule, but do not discuss them at all. What do they want to show with these numbers?

- The preceding comment is also true for table 4 about solvent accessibility of the divalent cations. What is the implication (if any) of this information?

- the free energy of binding of the divalent cations is always found to be very negative (favorable), even for Ca2 ion, which however leaves the protein during the simulation. I think that the numbers for Ca2 ion (and maybe also Zn2 and Ca3 ions) that experience lots of movements during the simulation are not reliable. I advise to compute these quantities over small “blocks” of 200ns and look if the values are converged.

- In section 2.4, the authors discuss the coordination geometry around the divalent cations. I think they should be more cautious about this because they are using a non-polarizable force field, which are not suited for that. They should at the very least warn the reader about the limitations of their methodology in that context.

Author Response

My main concern with this paper is that the objectives of the work are not really clearly defined. The authors used many tools to analyze their MD simulation. But it is hard to understand what they really want to show with these analyses and the relevance of their findings in a biological context, and compare them with available literature. For example, the authors cite three simulation papers on the same protein, but the comparison of their results with them is minimal. I think the authors should deepen their analysis to fully convince the reader of the relevance of their work. That is why I think the paper might be accepted for publication but only after major revision.We have added the following paragraph to the introduction clearly delineating the goals of the current study:The goal of the current study is to evaluate the dynamic stability of the divalent metal ions (2 Zn2+ and 3 Ca2+) reported in the X-ray crystal structure (PDB ID: 1CK7 and examine the domain movements within MMP-2 using 1.0 s NPT MD simulations at physiological temperature (310 K). Protein-metal cation distances and MMPBSA-interaction entropy binding energies (G) were calculated and the sampled conformational space analyzed with dihedral Principle Component Analysis (dPCA) and Dynamic Cross-Correlation Matrix (DCCM) analysis. The stability of the divalent ions and domain movement conformations will play an important role in the development of an additive force field model for protein·ligand docking studies, subsequent dynamic protein·ligand simulations and potential MMP-2 inhibitor development.The text of the results and discussion section has also been revised as outline in the red line revision provided to the editor making clear comparisons to the five previously published simulations (reference numbers given below) as well as highlighting important differences and relevant findings.15.   Díaz. N.; Suárez, D. Alternative Interdomain Configurations of the Full-Length MMP-2 Enzyme Explored by Molecular Dynamics Simulations. J Phys Chem B. 2012, 116(9), 2677-2686, DOI: 10.1021/jp211088d.30.   Díaz, N.; Suárez, D. Molecular Dynamics Simulations of Matrix Metalloproteinase 2: Role of the Structural Metal Ions. Biochem. 2007, 46(31), 8943-8952, DOI: 10.1021/bi700541p. 

31.   Díaz, N.; Suárez, D. Peptide Hydrolysis Catalyzed by Matrix Metalloproteinase 2: A computational Study. J Phys Chem B. 2008, 112(28), 8412-8424, DOI: 10.1021/jp803509h. 29.   Díaz N, Suárez D, Molecular dynamics simulations of the active matrix metalloproteinase-2: Positioning of the N-terminal fragment and binding of a small peptide substrate. Proteins. 2008, 72(1), 50-61, DOI: 10.1002/prot.21894. 14.   Díaz, N.; Suárez, D.; Valdés, H. From the X-ray Compact Structure to the Elongated form of the Full-Length MMP-2 Enzyme in Solution: A Molecular Dynamics Study. J Am Chem Soc. 2008, 130(43), 14070-14071, DOI: 10.1021/ja806090v.       There are long tables with lot of numbers (table 1 and table 6 especially). If the reader is not a specialist of the system, I guess he will not be able to decide what to look at, and the authors do not really help them in the text. Are these data really necessary in the main text? If so, I think the authors should spend more time in the text explaining which numbers are relevant and tell why. If not, maybe the tables could be shortened conserving only the most important numbers and putting all the other data as supplementary information.Tables 1 and 6 have been edited. We additionally have used the data and interaction criteria of Zheng et al. to define the coordination geometry limiting the analysis to those residues within the first valence shell of the metal ions.18.   Zheng, H.; Cooper D. R.; Porebski, P.J.; Shabalin, I. G.; Handing, K. B.; Minor, W. CheckMyMetal: a macromolecular metal-binding validation tool. Acta Cryst. 2017, D73, 223-233, DOI: 10.1107/S2059798317001061.

  17.   Zheng, H.; Chruszcz M.; Lasota P.; Lebioda, L.; Minor, M. Data mining of metal ion environments present in protein structures. J Inorg Biochem. 2008, 102(9), 1765-1776, DOI: 10.1016/j.jinorgbio.2008.05.006.   lines 88-92: the authors claim that the MD simulation has converged based on three quantities: configuration entropy, radius of gyration, and center of mass distance between domains. Obviously, looking at the two last measure, the simulation is not converged. A large variation is seen during the last 200ns. Maybe this is only a temporary conformational change, but it would be necessary to continue the simulations to establish it.The text has been revised to read:The Ca-trace configurational entropy of MMP-2 and associated divalent ions was calculated as a function of time (Figure S1) [19-21]. After a sharp rise in the configurational entropy over the first 100 ns, the value plateaus prior to 200 ns. Based on these results, we used the 200 ns to 1000 ns portion of the trajectory for our analysis with a sampling frequency of 0.1 ns. The radius of gyration (Rg) and inter-domain center-of mass (COM) distances: Cat-Hpx and Fib-Hpx, were analyzed with k-means clustering and the associated means and standard deviations of each population calculated (Figures S2 and S3) [22,23]. Due the large changes in Rg and COM distances between the domains and to insure that the size of the solvation box was adequate, the minimum distances between periodic images as a function of simulation time were determined using the g_mindist utility in GROMACS (periodic minimum distance: 3.81±0.75 nm). Five different distributions of protein conformations were identified with Rg: 2.65±0.02 nm, 2.70±0.06 nm, 2.80.±0.05 nm, 2.83±0.04 nm, and 3.09±0.13 nm. The COM distances mirror the Rg results, identifying five distributions of Cat-Hpx COM distances: 3.55±0.10 nm, 3.70±0.17 nm, 3.79±0.06 nm, 3.86±0.13 nm, and 4.20±0.23 nm. Five distributions of Fib-Hpx COM distances: 3.70±0.06 nm, 3.84±0.18 nm, 4.32±0.22 nm, 4.48±0.18 nm, and 5.52±0.48 nm, were also identified. The Rg and COM distance data are consistent with inter-domain motions between Cat/Fib and Hpx and the presence of inter-domain motions and the sampling of more extended conformations of MMP-2 in solution compared to the more compact X-ray crystal structure (PDB ID: 1CK7) which has Rg: 2.77 nm, Cat-Hpx COM distance: 3.81 nm, and Fib-Hpx COM distance: 2.00 nm [13]. These values are also consistent with those determined from prior simulations using the AMBER param03 force field [14,15]. It is not clear from either previous (multiple 200 ns simulations) or the current (1.0 ms) simulation if more extended conformations may be sampled or if an opening and closing of the Cat/Fib to Hpx domains occurs over longer time scales [14,15]

      line 83: I don’t know what “interaction entropy binding energies” means. I think the authors meant “binding free energies” that are actually computed with MM-PBSA.The following text has been added to the methods section:

The method defines the entropic contribution to the free energy term as:

(8)

where R is the universal gas constant, T the temperature, b is the thermodynamics beta:

(9)

b is Boltzmann’s constant, T is temperature, and DEplint is defined as:

(10)

⟨Eplint⟩ is the ensemble averaged protein-ligand interaction energy and Eplint is the protein-ligand interaction energy for a sampled conformation. For molecular dynamics simulations containing N sampled conformations, the interaction entropy can then be calculated as:

(11)

Duan, L.; Liu, X.; Zhang, J.Z.H. Interaction Entropy: A New Paradigm for Highly Efficient and Reliable Computation of Protein-Ligand Binding Free Energy, J Am Chem Soc. 2016, 138(17), 5722-5728, DOI: 10.1021/jacs.6b02682. Sun, A.; Yan, Y.N.; Yang, M.; Zhang, J.Z.H. Interaction entropy for protein-protein binding. J Chem Phys. 2017, 146(12), 124124 .DOI: 10.1063/1.4978893.

In table 2, authors give some numbers about RMSD of individual domains of the molecule, but do not discuss them at all. What do they want to show with these numbersThe following text has been added:The average Ca-trace RMSD data for MMP-2 as a whole and divided into its individual domains are given in Table 2. The most conformationally stable regions of the protein are the Cat and Hpx domains as observed in previous simulations with the majority of the Ca-trace flexibility accounted for within the three Fib domain, particularly the individual subdomains [14,15]. Although the RMSD of the Lnk region is low and comparable to that of Cat and Hpx, it should be considered that this in only a short 15 amino acid segment and that minor variations in the f/y dihedral angles can result in dramatic changes in the relationships between the Cat and Hpx domains [15].

      The preceding comment is also true for table 4 about solvent accessibility of the divalent cations. What is the implication (if any) of this information?The following text has been added to the Results and Discussion:

The radial distribution functions for the Zn2+ ions demonstrated peaks at 0.20 nm and 0.42 nm with troughs located at 0.31 nm and 0.50 nm. The Ca2+ ions has slight enlarged 1st and second hydration shells with peaks at 0.25 and 0.44 nm and troughs at 0.31 nm and 0.55 nm. The statistical densities of the first and second hydration shells (r(1st shell) and r(2nd shell)) are given in Table 4. These values represent the probabilities of a water molecule(s) occupying the volume of the hydration shell at any sampled time during the trajectory. There is only a minor difference in the 1st hydration shell comparing Zn2+ ion 1 and Zn2+ ion 2 with more significant changes in the 2nd hydration shell. This is consistent with Zn2+ ion 1 sequestered within the catalytic pocket of the enzyme while Zn2+ ion 2 dissociates from His178, His193, and His206 and interacting with Asp180 on the flexible S-loop. These results are also consistent with the solvent accessible surface area (SASA) which is much lower for Zn2+ ion 1 than Zn2+ ion 2. The results for the Ca2+ ions demonstrate that Ca2+ ion 1 is solvent sequestered with a low probability of water within the first hydration shell. Ca2+ ion 2 which dissociates from its binding site early in the simulation and appears to diffuse freely is similar the Ca2+ ion3 which stays in closer proximity to the Hpx domain but does not stay in close contact with its crystallographically associated residues. The SASA values in Table 4 also reflect this. The coordination numbers (CN) for the ions are given below. The value of 1 to 1.5 for Zn2+ ion 1 is consistent with prior studies [30,31]. The CN was also be calculated for Glu404 and is equal 1, in agreement with this residue’s catalytic importance and its proximity to the catalytic Zn2+ ion 1. The sequestered nature of Ca2+ ion 1 indicates only one associated water molecule while Ca2+ ion 2 and Ca2+ ion 3 are more normally hydrated.

Table 4 has been revised to include a detailed examination of the first and second hydration shells of the ions as well as their solvent accessible surface areas. The methods section has also been revised to read:

The distances from the ion to the first and second hydration shells is determined by examining the peaks and troughs of the associated RDF. The integral of the RDF (g(r)):

(5)

is the statistical density of the surrounding solvent and represents the probability of a solvent molecule being located within r distance of the central atom or group at any sampled time. At infinite distance this probability is by definition 1.0. The coordination number can then be calculated as:

(6)

where r is the density and r the distance from the ion [52,53].

The free energy of binding of the divalent cations is always found to be very negative (favorable), even for Ca2 ion, which however leaves the protein during the simulation. I think that the numbers for Ca2 ion (and maybe also Zn2 and Ca3 ions) that experience lots of movements during the simulation are not reliable. I advise to compute these quantities over small “blocks” of 200ns and look if the values are converged.The protein-metal ion interaction energies were calculated for the for the entire run and found to plateau at the same rate and time period as the Ca- trace configurational entropy. It is also important to note that the Mobile ions (Zn2+ ion 2, Ca2+ ion 2, and Ca2+ ion 3) rapidly move outside of their crystallographically defined binding pockets, within 200 ns of the start of the simulation.

    In section 2.4, the authors discuss the coordination geometry around the divalent cations. I think they should be more cautious about this because they are using a non-polarizable force field, which are not suited for that. They should at the very least warn the reader about the limitations of their methodology in that context.We agree with this point and have revised our analysis based on references 17 and 18 as discussed answering Question #2. WE have also add the following text to the conclusion section:

The current non-bonded model of Li et al. with the CHARMM36m force field appears to be a reasonable model of protein metal cation interactions in MMP-2.The data suggest that further models should retain Zn2+ ion 1, Zn2+ ion 2 and Ca2+ ion 1 since they play important chemical and conformational roles and remove Ca2+ ion 2 and possibly Ca2+ ion 3 since their role is minimal. Protein·ligand binding studies should also include multiple conformations to account for the mobility of the Hpx domain with respect to the Cat and Fib domains. We do however acknowledge that the Li et al. with the CHARMM36m force field has known limitations. Accurate protein·metal ion interaction energies and coordination geometries require either the use of a carefully parameterized polarizable force field or more computationally expensive QM/MM simulations.

Reviewer 2 Report

The paper of Voit-Ostricki et al. provides interesting results on the stability of Human Matrix Metalloproteinase-2 in the presence of Zn2+ and Ca2+ ions The set-up of the in silico test is presented in great details and the results are discussed in a straightforward manner. Anyway, the authors should revise their work as follows:

Line 235: It is not clear what the authors meant by “ … the X-ray l structure …”. Please reconsider or provide more details.

Some references are not correctly quoted in the manuscript. Take for instance Li et al. (line 251), Diaz et al. (line 337)

Although common, the meaning of the parameters and variables used in equations 2-4 should be provided.

The Conclusions section should be reconsidered. No literature should be quoted in the Conclusions section. The authors should present only the main findings of their own study. 

Author Response

Line 235: It is not clear what the authors meant by “ … the X-ray l structure …”. Please reconsider or provide more details.The line of text has been corrected to read:Initial coordinates were obtained from the X-ray crystal structure of the Glu404 to Ala404 mutant of the human MMP-2 (PDB ID: 1CK7) [13].

      Some references are not correctly quoted in the manuscript. Take for instance Li et al. (line 251), Diaz et al. (line 337)..We have reviewed the references and made the necessary corrections. The relevance and associations of the cited reference numbers (36 through 41 as cited above, now references 39 through 44) from line 251 of the original manuscript are as follows:39.       Huang, J.; Rauscher, S.; Nawrocki, G.; Ran, T.; Feig, M.; de Groot, B.L.; Grubmüller, H.; MacKerell, A.D. Jr. CHARMM36m: an improved force field for folded and intrinsically disordered proteins, Nat Methods. 2017, 14(1), 71-73, DOI: 10.1038/nmeth.4067.40.       Best, R.B.; Zhu, X.; Shim, J.; Lopes, P.E.; Mittal, J.; Feig, M.; Mackerell, A.D. Jr. Optimization of the Additive CHARMM All-Atom Protein Force Field Targeting Improved Sampling of the Backbone ,  and Side-Chain 1 and 2 Dihedral Angles, J Chem Theory Comput. 2012, 8(9), 3257-3273. DOI: 10.1021/ct300400xC.    The general CHARMM forcefield or proteins.D.    The implementation of CHARMM36 in GROMACSE.    The Li 2+ metal ion parameters. 

44.       Li, P. Roberts, B.P.; Chakravorty, D.K.; Merz, K.M. Jr. Rational Design of Particle Mesh Ewald Compatible Lennard-Jones Parameters for +2 Metal Cations in Explicit Solvent. J. Chem Theory Comput. 2013, 9(6), 2733-2748, DOI: 10.1021/ct400146w. 43.       Bjelkmar, P.; Larsson, P.; Cuendet, M.A.; Bess, B.; Lindahl, E. Implementation of the CHARMM force field in GROMACS: Analysis of protein stability effects from correction maps, virtual interaction sites, and water models, J Chem Theory Comput. 2010, 6(2), 459-466, DOI: 10.1021/ct900549r. 42.       MacKerell, A.D. Jr.; Bashford, D.; Bellott, M.; Dunbrack, R.L.; Evanseck, J.D.; Field, M.J.; Fischer, S.; Gao, J.; Guo, H.; Ha, S.; Joseph-McCarthy, D.; Kuchnir, L.; Kuczera, K.; Lau, F.T.; Mattos, C.; Michnick, S.; Ngo, T.; Nguyen, D.T.; Prodhom, B.; Reiher, W.E.; Roux, B.; Schlenkrich, M.; Smith, J.C.; Stote, R.; Straub, J.; Watanabe, M.; Wiórkiewicz-Kuczera, J.; Yin, D.; Karplus, M. All-atom empirical potential for molecular modeling and dynamics Studies of proteins, J. Phys. Chem. B 1998, 102(18), 3586-3616, DOI: 10.1021/jp973084f. 41.       MacKerell, A.D. Jr.; Feig, M.; Brooks, C.L. III. Extending the treatment of backbone energetics in protein force fields: limitations of gas-phase quantum mechanics in reproducing protein conformational distributions in molecular dynamics simulations, J Comp Chem. 2004, 25(11), 1400-1415, DOI: 10.1002/jcc.20065 B.    The original CHARMM36 force field development A.    CHARMM36m and TIP3Pm   Although common, the meaning of the parameters and variables used in equations 2-4 should be provided.The following has been added to the methods section to clarify meaning of the parameters: sij, eij, qi, qj, rij, and e0 in equations 2 through 4:

The metal ions are represented with a standard Lennard-Jones and coulomb potential energy functions [45]:

(2)

where, sij, is the distance between two atoms at their lowest potential energy calculated as an arithmetic mean,

(3)

eij, is the depth of the potential energy well calculated as a geometric mean,

(4)

qi and qj are the respective point charges, rij is the distance separating the two atoms, e0 is the dielectric constant ,and i and j are atom indices.

The Conclusions section should be reconsidered. No literature should be quoted in the Conclusions section. The authors should present only the main findings of their own study.The conclusion section has been revised and literature references removed. The current section reads as follows:

We report a microsecond scale molecular dynamics analysis of the full biologically active protein (Cat, Fib, and Hpx domains with Lnk region) with its crystallographically associated (structural) divalent metal ions (Zn2+ and Ca2+). Our model indicates that MMP-2 undergoes significant inter domain motions. The Rg and COM data demonstrate five macro distributions of conformations based on size. The dPCA data is consistent with 10 families of conformations. The three lowest energy populations of conformations differ predominantly in the orientation of the Hpx domain in relation to the Cat and Fib domains. This is confirmed by the DCCM analysis where the difference in orientation of the Hpx to the Cat and Fib domains comprises the first two principle components. These inter domain movements are facilitated by the flexible linker region Gly446 through Asp476 and may play and important role in collagen substrate binding, unravelling and subsequent catalysis

Zn2+ ion 1 (catalytic ion) and Ca2+ ion 1 are tightly bound within their crystallographically defined pockets. Zn2+ ion 2 was more flexible and associated with the S-loop which has demonstrated increased flexibility in both prior simulations and our own however, the use of short simulation time (10 ns) and a harmonic potential representing the interaction between the Zn2+ ion and the associated His and Asp/Glu residues may have artificially stabilized the protein-Zn2+ ion 2 interaction. Ca2+ ion 2 and Ca2+ ion 3 do not remain in their crystallographically defined positions. Although Ca2+ ion 3 is more closely associated with the Hpx domain, the carboxyl sidechains of the Asp residues that contribute to its interaction are at distances well outside of the coordination sphere.

The current non-bonded model of Li et al. with the CHARMM36m force field appears to be a reasonable model of protein metal cation interactions in MMP-2.The data suggest that further models should retain Zn2+ ion 1, Zn2+ ion 2 and Ca2+ ion 1 since they play important chemical and conformational roles and remove Ca2+ ion 2 and possibly Ca2+ ion 3 since their role is minimal. Protein·ligand binding studies should also include multiple conformations to account for the mobility of the Hpx domain with respect to the Cat and Fib domains. We do however acknowledge that the Li et al. with the CHARMM36m force field has known limitations. Accurate protein·metal ion interaction energies and coordination geometries require either the use of a carefully parameterized polarizable force field or more computationally expensive QM/MM simulations.

Reviewer 3 Report

Summary: In this manuscript, Voit-Ostricki and co-workers perform molecular dynamics simulations on human MMP2 in the presence of Ca2+ and Zn2+. They carefully examine the metal binding interactions between MMP2 and the metals as well as the movement of the structural motifs and domains over the simulation time of 1 microsecond. They find that Ca2+ ion 2 seems to be less stably bound to the MMP2 compared to Zn2+ ions 1 and 2 and Ca2+ ions 1 and 3. Their structural analysis revealed that the LNK region connecting the catalytic and hemopexin domains.

Overall, I have only minor suggestions for improvement on the MD simulations execution and analysis. My biggest concern is that I don’t believe in the present form the manuscript increases our understanding of how MMP2 proteins function. That information may be in the data, but the authors have done a poor job in articulating the importance of their finding. Thus, I don’t recommend publication in the current form. If the authors can using their MD data to provide compelling new insights into the mechanism of MMP2 action, I think the paper may garner some interest.

My other concerns are listed below:

Figure 1 – side chain labels are small and difficult to read.

Lines 66-69 seem like they should be part of a figure legend, not part of the main text.

Table 1 – Some explanation on what the optimum (most stable) coordination geometry is for Zn2+ and Ca2+ is needed.

Figure 2- The authors show the DSSP assigned secondary structured. The figure legend states (“Sampled >50%)”. Do the authors mean “sampled >50% of the simulation time”? Clarification is needed – particularly since this was determined with an in-house script.

How do the authors know that their water simulation box is large enough to allow an entire range of domain movements possible by the protein?

What concentration of Zn2+ and Ca2+ were in the simulation box? How do these concentrations relate to physiological levels?

Figure 4 – I think this figure may be more informative if the authors structurally align the backbone atoms of only the secondary structure components. Additionally, an alignment of all domains (again using the backbone atoms only of the secondary structure components) except the Fib domain would better illustrate the motion of the Fib domain.

Line 154 – The authors mention that Zn2+ ion 1 is “hydrated” with an H2O molecule that is 0.5 nm away; but this distance seems too far from Zn2+ to be hydrating it. Further, Table 4 does not really suggest to me any increased probability of hydration of Zn2+ ion 1 compared to the other ions. The table needs to be better annotated to show that Zn2+ hydration probability is greater than the other ions.

The color coding in Figure 5 and 6 is confusing. I would try to be consistent in both (i.e. red for correlation and blue for anti-correlation) in both figures.

Table 5 – binding energies should also be converted into equilibrium constants and compared with experimentally measured values.

It seems that Zn2+ binding to His residues would be highly pH dependent. What happens at lower and higher pH? In the methods, pH is set to 7; however, this pH is very close to the pKa of His (so 50% of His would be charged and 50% uncharged). Is this charge equilibrium of His at pH ~7 taken into account by the simulations?

Lines 199-201 – the authors suggest that “These residues (Asp and Glu residues within the Cat domain) contribute significantly to MMP-2 to  Zn2+ ion 1 interaction energy despite being outside what is considered to be the normal coordination sphere of the ion.” The authors should explain how this may be possible.

Line 209 – the authors state that the “…with the difference attributed to increased flexibility of the Glu residue secondary to presence of the extra methyl group.” Should be “methylene” group.

Author Response

Overall, I have only minor suggestions for improvement on the MD simulations execution and analysis. My biggest concern is that I don’t believe in the present form the manuscript increases our understanding of how MMP2 proteins function. That information may be in the data, but the authors have done a poor job in articulating the importance of their finding. Thus, I don’t recommend publication in the current form. If the authors can using their MD data to provide compelling new insights into the mechanism of MMP2 action, I think the paper may garner some interest.We have added the following paragraph to the introduction clearly delineating the goals of the current study:The goal of the current study is to evaluate the dynamic stability of the divalent metal ions (2 Zn2+ and 3 Ca2+) reported in the X-ray crystal structure (PDB ID: 1CK7 and examine the domain movements within MMP-2 using 1.0 s NPT MD simulations at physiological temperature (310 K). Protein-metal cation distances and MMPBSA-interaction entropy binding energies (G) were calculated and the sampled conformational space analyzed with dihedral Principle Component Analysis (dPCA) and Dynamic Cross-Correlation Matrix (DCCM) analysis. The stability of the divalent ions and domain movement conformations will play an important role in the development of an additive force field model for protein·ligand docking studies, subsequent dynamic protein·ligand simulations and potential MMP-2 inhibitor development.The text of the results and discussion section has also been revised as outline in the red line revision provided to the editor making clear comparisons to the five previously published simulations (reference numbers given below) as well as highlighting important differences and relevant findings.15.   Díaz. N.; Suárez, D. Alternative Interdomain Configurations of the Full-Length MMP-2 Enzyme Explored by Molecular Dynamics Simulations. J Phys Chem B. 2012, 116(9), 2677-2686, DOI: 10.1021/jp211088d.30.   Díaz, N.; Suárez, D. Molecular Dynamics Simulations of Matrix Metalloproteinase 2: Role of the Structural Metal Ions. Biochem. 2007, 46(31), 8943-8952, DOI: 10.1021/bi700541p. 

31.   Díaz, N.; Suárez, D. Peptide Hydrolysis Catalyzed by Matrix Metalloproteinase 2: A computational Study. J Phys Chem B. 2008, 112(28), 8412-8424, DOI: 10.1021/jp803509h. 29.   Díaz N, Suárez D, Molecular dynamics simulations of the active matrix metalloproteinase-2: Positioning of the N-terminal fragment and binding of a small peptide substrate. Proteins. 2008, 72(1), 50-61, DOI: 10.1002/prot.21894. 14.   Díaz, N.; Suárez, D.; Valdés, H. From the X-ray Compact Structure to the Elongated form of the Full-Length MMP-2 Enzyme in Solution: A Molecular Dynamics Study. J Am Chem Soc. 2008, 130(43), 14070-14071, DOI: 10.1021/ja806090v.       Figure 1 – side chain labels are small and difficult to read.The side chain labels in Figure 1 have been increased in size to improve readability.

    Lines 66-69 seem like they should be part of a figure legend, not part of the main textThis text has been removed as suggested.

    Figure 2- The authors show the DSSP assigned secondary structured. The figure legend states (“Sampled >50%)”. Do the authors mean “sampled >50% of the simulation time”? Clarification is needed – particularly since this was determined with an in-house script.The text has been revised to read:The DSSP assigned secondary structure (sampled >50% of the simulation time) is shown at the top of the graph (blue, a-helix; red, b-sheet; green, b-turn/bend; and aqua, coil). The Cat, Fib and Hpx domains and Lnk regions are demarcated with dotted lines.

      How do the authors know that their water simulation box is large enough to allow an entire range of domain movements possible by the protein?The following text has been added to the Results and Discussion section:Due the large changes in Rg and COM distances between the domains and to insure that the size of the solvation box was adequate, the minimum distances between periodic images as a function of simulation time were determined using the g_mindist utility in GROMACS (periodic minimum distance: 3.81±0.75 nm).

      What concentration of Zn2+ and Ca2+ were in the simulation box? How do these concentrations relate to physiological levels?The concentration of MMP-2 within the solvation box was 1.1 mM. The concentration of the bound Zn2+ and Ca2+ ions were 2.2 mM and 3.3 mM respectively (two Zn2+ and three Ca2+ ions per protein molecule). The concentration of ”free” (unbound Zn2+ and Ca2+) in human plasma is approximately 17 mM and 1.5 mM respectively according to Documenta Geigy Scientific Tables. The goal of the current simulation is to investigate the stability of the X-ray crystal structure bound Zn2+ and Ca2+ and their effects on MMP-2 conformation while maintaining direct comparability to previous simulations by Diaz et al. The following text has been added to the methods section:The size of the box was 1458.26 nm3, containing 8487 protein and divalent metal cation atoms and 45106 waters giving an MMP-2 concentration of 1.1 mM, a Zn2+ concentration of 2.2 mM and a Ca2+ concentration of 3.3 mM.

      Figure 4 – I think this figure may be more informative if the authors structurally align the backbone atoms of only the secondary structure components. Additionally, an alignment of all domains (again using the backbone atoms only of the secondary structure components) except the Fib domain would better illustrate the motion of the Fib domain.Prior to submission, we experimented with a variety of different overlay schemes to generate Figure 4. The provided image is the best quality one obtained from that work.

    Line 154 – The authors mention that Zn2+ ion 1 is “hydrated” with an H2O molecule that is 0.5 nm away; but this distance seems too far from Zn2+ to be hydrating it. Further, Table 4 does not really suggest to me any increased probability of hydration of Zn2+ ion 1 compared to the other ions. The table needs to be better annotated to show that Zn2+ hydration probability is greater than the other ionsThe following text has been added to the Results and Discussion:

The radial distribution functions for the Zn2+ ions demonstrated peaks at 0.20 nm and 0.42 nm with troughs located at 0.31 nm and 0.50 nm. The Ca2+ ions has slight enlarged 1st and second hydration shells with peaks at 0.25 and 0.44 nm and troughs at 0.31 nm and 0.55 nm. The statistical densities of the first and second hydration shells (r(1st shell) and r(2nd shell)) are given in Table 4. These values represent the probabilities of a water molecule(s) occupying the volume of the hydration shell at any sampled time during the trajectory. There is only a minor difference in the 1st hydration shell comparing Zn2+ ion 1 and Zn2+ ion 2 with more significant changes in the 2nd hydration shell. This is consistent with Zn2+ ion 1 sequestered within the catalytic pocket of the enzyme while Zn2+ ion 2 dissociates from His178, His193, and His206 and interacting with Asp180 on the flexible S-loop. These results are also consistent with the solvent accessible surface area (SASA) which is much lower for Zn2+ ion 1 than Zn2+ ion 2. The results for the Ca2+ ions demonstrate that Ca2+ ion 1 is solvent sequestered with a low probability of water within the first hydration shell. Ca2+ ion 2 which dissociates from its binding site early in the simulation and appears to diffuse freely is similar the Ca2+ ion3 which stays in closer proximity to the Hpx domain but does not stay in close contact with its crystallographically associated residues. The SASA values in Table 4 also reflect this. The coordination numbers (CN) for the ions are given below. The value of 1 to 1.5 for Zn2+ ion 1 is consistent with prior studies [30,31]. The CN was also be calculated for Glu404 and is equal 1, in agreement with this residue’s catalytic importance and its proximity to the catalytic Zn2+ ion 1. The sequestered nature of Ca2+ ion 1 indicates only one associated water molecule while Ca2+ ion 2 and Ca2+ ion 3 are more normally hydrated.

Table 4 has been revised to include a detailed examination of the first and second hydration shells of the ions as well as their solvent accessible surface areas. The methods section has also been revised to read:

The distances from the ion to the first and second hydration shells is determined by examining the peaks and troughs of the associated RDF. The integral of the RDF (g(r)):

(5)

is the statistical density of the surrounding solvent and represents the probability of a solvent molecule being located within r distance of the central atom or group at any sampled time. At infinite distance this probability is by definition 1.0. The coordination number can then be calculated as:

(6)

where r is the density and r the distance from the ion [52,53].

The color coding in Figure 5 and 6 is confusing. I would try to be consistent in both (i.e. red for correlation and blue for anti-correlation) in both figuresFigures 5 and 6 have been edited so that the color bar used to indicate correlation and anti-correlation are the same.

    Table 5 – binding energies should also be converted into equilibrium constants and compared with experimentally measured valuesAlthough the stoichiometry of MMP-2 has been defined, there are no published equilibrium constants for Zn2+ and/or Ca2+ interacting with MMP-2 making a comparison to experimentally measured values impossible.We would also emphasize the following as outlined in our conclusions:We do however acknowledge that the Li et al. with the CHARMM36m force field has known limitations. Accurate protein·metal ion interaction energies and coordination geometries require either the use of a carefully parameterized polarizable force field or more computationally expensive QM/MM simulations.

    33.  Springman, E.B.; Nagase, H.; Birkedal-Hansen, H.; Van Wart; H.E. Zinc content and function in human fibroblast collagenase. Biochemistry 1995; 34: 15713-15720. DOI: 10.1021/bi00048a016   It seems that Zn2+ binding to His residues would be highly pH dependent. What happens at lower and higher pH? In the methods, pH is set to 7; however, this pH is very close to the pKa of His (so 50% of His would be charged and 50% uncharged). Is this charge equilibrium of His at pH ~7 taken into account by the simulations?Prior NMR studies have demonstrated that the pKa of His for the protonation of the neurtal t tautomer to the biprotonated state is 6.0. The following text and reference has been added to the methods section:The protonation state and charges of all other residues within the protein were set to correspond to a pH of 7.0 with the His residues sidechains assigned in their uncharged state consistent with NMR titration studies [38].38.  Li, S.; Hong, M. Protonation, Tautomerization, and Rotameric Structure of Histidine: A comprehensive Study by Magic-Angle-Spinning Solid-State NMR. J Am Chem Soc. 2011, 133(5), 1534-1544, DOI: 10.1021/ja108943n.

        Lines 199-201 – the authors suggest that “These residues (Asp and Glu residues within the Cat domain) contribute significantly to MMP-2 to Zn2+ ion 1 interaction energy despite being outside what is considered to be the normal coordination sphere of the ion.” The authors should explain how this may be possible.The analysis has been revised and restricted to only those residues and atoms within the first coordination sphere of the divalent metal ion. We have also added the following text to the results and discussion emphasizing the long range coulombic forces involved:

Ca2+ ion 1 has strong interactions with the adjacent Od atoms of Asp and Oe atoms of Glu of adjacent residues. There is a shift from the divalent cation to backbone carbonyl oxygen interactions that are observed in the crystal structure to interactions dominated by the acidic sidechain groups. The coordination geometry changes from pentagonal pyramidal to a square planar geometry. The strong interactions and coordination with these sidechains is expected and has been previously observed for other systems [34]. The results for Ca2+ ion 1 are the same as noted in prior simulations with a shift from backbone carbonyl interactions which may represent crystal packing forces to interactions with the carboxyl sidechains [30]. In general the favorability of interaction is Glu>Asp with the difference attributed to increased flexibility of the Glu residue secondary to presence of the extra methylene group. The binding of this ion is also similar to that of the catalytic Zn2+ ion 1 in that adjacent but non-coordinated electronegative Asp and Glu residues make significant contributions to its binding energy secondary to the strong long range coulomb interactions. Ca2+ ion 2 and Ca2+ ion 3 represent special cases. Ca2+ ion 2 freely diffuses out of its binding site and any strong interactions with electrostatic sidechains are transient. The x-ray crystal structure associated atoms are given to illustrate this, Although prior simulations do note the Ca2+ ion 2 leaving its binding site, they do note that the associated interactions are weak and the study is most likely limited by the short simulation time (10 ns) [30]. Ca2+ ion 3 is more stable in its RMSF compared to Ca2+ ion 2 (Table 3) however, it is still much more variable that the other associated ions. This ion breaks its association with the backbone carbonyl atoms of Asp476, Asp 521, Asp569 and Asp618 to form strong interactions with multiple Asp carboxyl oxygens as given in Table 6. This result is most likely due to the change from crystal packing forces to that of an aqueous environment [34].

Line 209 – the authors state that the “…with the difference attributed to increased flexibility of the Glu residue secondary to presence of the extra methyl group.” Should be “methylene” group.The text has been corrected from methyl to methylene group.

Round 2

Reviewer 3 Report

Overall, the authors have done a good job addressing my concerns. A couple of minor points of disagreement that I have with a couple of responses are listed below. I believe that the manuscript would be improved if the authors could address these points prior to publications:

1) In the response to point 6 – the authors argue that free Zn2+ concentration is 17 mM (millimolar). This is highly over-estimated and is likely referring to total Zn2+. This level of Zn2+ would cause widespread protein precipitation and be highly toxic to cells. Free Zn2+ is likely in the nM range (nanomolar). While the authors have added text in the manuscript stating what the concentration of Zn2+ and Ca2+ is in their simulation box, they have not added any text on the free physiological concentrations of these ions (as I originally suggested). I suggest a sentence and the appropriate referenced are added for these free physiological Zn2+ and Ca2+ concentrations.

2) In response to point 11 – the authors argue that the pKa for His is 6.0; however, the pKa of His incorporated into protein can vary compared to free His and depending on the protein. I suggest a sentence in the Discussion stating that the protonation state of the His residues could affect Zn2+ binding, and a limitation of the present study is not knowing and modeling the His protonation states of MMP-2.

Author Response

In the response to point 6 – the authors argue that free Zn2+ concentration is 17 mM (millimolar). This is highly over-estimated and is likely referring to total Zn2+. This level of Zn2+ would cause widespread protein precipitation and be highly toxic to cells. Free Zn2+ is likely in the nM range (nanomolar). While the authors have added text in the manuscript stating what the concentration of Zn2+ and Ca2+ is in their simulation box, they have not added any text on the free physiological concentrations of these ions (as I originally suggested). I suggest a sentence and the appropriate referenced are added for these free physiological Zn2+ and Ca2+ concentrations.   The following text and references have been added to the Materials and Methods section:   47.  Foote, J.W.; Delves, H.T. Determination of non-protein-bound zinc in human serum using ultrafiltration and atomic absorption spectrometry with electrothermal atomisation. Analyst. 1988, 113(6), 911-915, DOI: 10.1039/AN9881300911.  48.  Forman, D.T.; Lorenzo, L. Ionized Calcium: Its significance and Clinical Usefulness. Ann Clin Lab Sci. 1991, 21(5), 297-304.   The mean ± standard deviation concentrations of non-protein bound Zn2+ and Ca2+ in the plasma of healthy human adults are: 81 ± 16 nM and 1.19 ± 0.06 mM respectively [47,48].

In response to point 11 – the authors argue that the pKa for His is 6.0; however, the pKa of His incorporated into protein can vary compared to free His and depending on the protein. I suggest a sentence in the Discussion stating that the protonation state of the His residues could affect Zn2+ binding, and a limitation of the present study is not knowing and modeling the His protonation states of MMP-2.   The following text and reference (reference 34 was previously reference 38) have been added to the Results and Discussion section: 35.  Langella, E.; Improta, R.; Barone, V. Checking the pH-Induced Conformational Transition of Prion Protein by Molecular Dynamics Simulations: Effect of Protonation of Histidine Residues. Biophys J. 2004, 87(6), 3623-3632, DOI: 10.1529/biophysj.104.043448.   It should be noted that the current study assumes that all His residue sidechains are uncharged with a pKa = 6.0 at a system pH = 7.0 [34]. This assumption cannot account for local environments that may cause the His sidechains to be protonated effecting Zn2+ binding as has been demonstrated for His residue interactions with other divalent cations and their metal binding site [35].